# PDE-REGULARIZED NEURAL NETWORKS FOR IMAGE CLASSIFICATION

## ABSTRACT

Neural ordinary differential equations (neural ODEs) introduced an approach to approximate a neural network as a system of ODEs after considering its layer as a continuous variable and discretizing its hidden dimension. While having several good characteristics, neural ODEs are known to be numerically unstable and slow in solving their integral problems, resulting in errors and/or much computation of the forward-pass inference. In this work, we present a novel partial differential equation (PDE)-based approach that removes the necessity of solving integral problems and considers both the layer and the hidden dimension as continuous variables. Owing to the recent advancement of learning PDEs, the presented novel concept, called PR-Net, can be implemented. Our method shows comparable (or better) accuracy and robustness in much shorter forward-pass inference time for various datasets and tasks in comparison with neural ODEs and Isometric MobileNet V3. For the efficient nature of PR-Net, it is suitable to be deployed in resource-scarce environments, e.g., deploying instead of MobileNet.

## 1 INTRODUCTION

It had been discovered that interpreting neural networks as differential equations is possible by several independent research groups (Weinan, 2017; Ruthotto & Haber, 2019; Lu et al., 2018; Ciccone et al., 2018; Chen et al., 2018; Gholami et al., 2019). Among them, the seminal neural ordinary differential equation (neural ODE) research work, which considers the general architecture in Figure 1 (a), is to learn a neural network approximating $\frac{\partial \boldsymbol{h}(t)}{\partial t}$, where $\boldsymbol{h}(t)$ is a hidden vector at layer (or time) $t$ (Chen et al., 2018). As such, a neural network is described by a system of ODEs, each ODE of which describes a dynamics of a hidden element. While neural ODEs have many good characteristics, they also have limitations, which are listed as follows:

**Pros.** Neural ODEs can interpret $t$ as a continuous variable and we can have hidden vectors at any layer (or time) $l$ by $\boldsymbol{h}(l) = \boldsymbol{h}(0) + \int_0^l o(\boldsymbol{h}(t), t; \boldsymbol{\theta}_o)\, dt$, where $o(\boldsymbol{h}(t), t; \boldsymbol{\theta}_o) = \frac{\partial \boldsymbol{h}(t)}{\partial t}$ is a neural network parameterized by $\boldsymbol{\theta}_o$.

**Pros.** Neural ODEs sometimes have smaller numbers of parameters than those of other conventional neural network designs, e.g., (Pinckaers & Litjens, 2019).

**Cons.** Neural ODEs, which use an adaptive step-size ODE solver, sometimes show numerical instability (i.e., the underflow error of the step-size) or their forward-pass inference can take a long time (i.e., too many steps) in solving integral problems, e.g, a forward-pass time of 37.6 seconds of ODE-Net vs. 9.8 seconds of PR-Net in Table 2. Several countermeasures have been proposed but it is unavoidable to solve integral problems (Zhuang et al., 2020; Finlay et al., 2020; Daulbaev et al., 2020).

To tackle the limitation, we propose the concept of partial differential equation-regularized neural network (PR-Net) to directly learn a hidden element, denoted $h(d, t)$ at layer (or time) $t \in [0, T]$ and dimension $d \in \mathbb{R}^m$. Under general contexts, a PDE consists of i) an initial condition at $t = 0$, ii) a boundary condition at a boundary location of the spatial domain $\mathbb{R}^m$, and iii) a governing equation describing $\frac{\partial h(d,t)}{\partial t}$. As such, learning a PDE from data can be reduced to a regression-like problem to predict $h(d, t)$ that meets its initial/boundary conditions and governing equation.

In training our proposed PR-Net, $\boldsymbol{h}(0)$ is provided by an earlier feature extraction layer, which is the same as neural ODEs. However, an appropriate governing equation is unknown for downstream

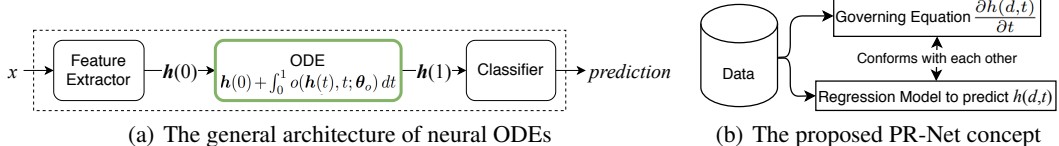

(a) The general architecture of neural ODEs      (b) The proposed PR-Net concept

Figure 1: The proposed PR-Net avoids solving integral problems by learning a regression model that conforms with a learned governing equation.

machine learning tasks. Therefore, we propose to train a regression model for predicting $h(d, t)$ and its governing equation simultaneously (see Figure 1 (b)). In other words, neural ODEs directly learn a governing equation (i.e., $\frac{\partial \boldsymbol{h}(t)}{\partial t}$), whereas PR-Net learns a governing equation in conjunction with a regression model that conforms with the learned governing equation. The key advantage in our approach is that we can eliminate the necessity of solving integral problems — in neural ODEs, where we learn a governing equation only, solving integral problems is mandatory.

Such forward and inverse problems (i.e., solving PDEs for $h(d, t)$ and identifying governing equations, respectively) arise in many important computational science problems and there have been many efforts applying machine learning/deep learning techniques to those problems (e.g., in earth science (Reichstein et al., 2019; Bergen et al., 2019) and climate science (Rolnick et al., 2019)). Recently, physics-informed or physics-aware approaches (Battaglia et al., 2016; Chang et al., 2017; de Bezenac et al., 2018; Raissi et al., 2019; Sanchez-Gonzalez et al., 2018; Long et al., 2018) have demonstrated that designing neural networks to incorporate prior scientific knowledge (e.g., by enforcing physical laws described in governing equations (Raissi et al., 2019)) greatly helps avoiding over-fitting and improving generalizability of the neural networks. There also exist several approaches to incorporate various ideas of classical mechanics in designing neural-ODE-type networks (Greydanus et al., 2019; Chen et al., 2020; Cranmer et al., 2020; Zhong et al., 2020; Lee & Parish, 2020). However, all these works are interested in solving either forward or inverse problems whereas we solve the two different problem types at the same time for downstream tasks. The most similar existing work to our work is in (Long et al., 2018). However, this work studied scientific PDEs and do not consider $t$ as a continuous variable but use a set of discretized points of $t$.

Compared to previous approaches, the proposed method has a distinct feature that forward and inverse problems are solved simultaneously with a continuous variable $t$. Due to this unique feature, the method can be applied to general machine learning downstream tasks, where we do not have a priori knowledge on governing equations, such as image classification. Our proposed PR-Net had the following characteristics:

**Pros.** PR-Net trains a regression model that outputs a scalar element $h(d, t)$ (without solving any integral problems), and we can consider both $d$ and $t$ as continuous variables. Therefore, it is possible to construct flexible hidden dimension vectors.

**Pros.** PR-Net does not require solving integral problems. As such, there is no numerical instability and their forward-pass time is much shorter than that of neural ODEs.

**Pros.** By learning a governing equation, we can regularize the overall behavior of PR-Net.

**Cons.** PR-Net sometimes requires a larger number of parameters than that of neural ODEs or conventional neural networks.

## 2   PARTIAL DIFFERENTIAL EQUATIONS

The key difference between ODEs and PDEs is that PDEs can have derivatives of multiple variables whereas ODEs should have only one such variable's derivative. Therefore, our PDE-based method interprets both the layer of neural network and the dimension of hidden vector as continuous variables, which cannot be done in neural ODEs. In our context, $h(d, t)$ means a hidden scalar element at layer $t \in \mathbb{R}$ and dimension $d \in \mathbb{R}^m$, e.g., $m = 1$ if $\boldsymbol{h}(t)$ is a vector, $m = 3$ if $\boldsymbol{h}(t)$ is a convolutional feature map, and so on.

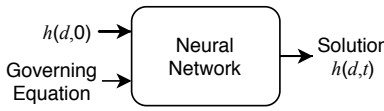

Figure 2: A neural network predicts solution values at $d, t$ given initial conditions, denoted $h(d, 0)$ for various $d$, and a governing equation.

Table 1: Two types of PDE problems related to our work

| Type | Data | What to infer |
|---|---|---|
| Forward Problem | - Initial condition
- Governing equation | Solution $h(d, t)$ |
| Inverse Problem | - Solution $h(d, t)$
- Initial condition | Governing equation |

In this section, we first introduce the forward and inverse problems of PDEs in general contexts (see Table 1). Then, we extend them to design our proposed method in deep-learning contexts.

## 2.1 FORWARD PROBLEM OF PDEs IN GENERAL CONTEXTS

The forward PDE problem in general contexts is to find a solution $h(d, t)$, where $d$ is in a spatial domain $\mathbb{R}^m$ and $t$ is in a time domain $[0, T]$, given i) an initial condition $h(d, 0)$, ii) a boundary condition $h(d_{bc}, t)$, where $d_{bc}$ is a boundary location of the spatial domain $\mathbb{R}^m$, and iii) a governing equation $g$ (Raissi et al., 2019) We note that the boundary condition can be missing in some cases (Kim, 2018). The governing equation is typically in the following form with particular choices of $\alpha_{i,j}$ (Raissi, 2018; Peng et al., 2020):

$$
\begin{aligned}
g(d, t; h) \stackrel{\text{def}}{=} h_t - \big( & \alpha_{0,0} + \alpha_{1,0}h + \alpha_{2,0}h^2 + \alpha_{3,0}h^3 \\
& + \alpha_{0,1}h_d + \alpha_{1,1}hh_d + \alpha_{2,1}h^2h_d + \alpha_{3,1}h^3h_d \\
& + \alpha_{0,2}h_{dd} + \alpha_{1,2}hh_{dd} + \alpha_{2,2}h^2h_{dd} + \alpha_{3,2}h^3h_{dd} \\
& + \alpha_{0,3}h_{ddd} + \alpha_{1,3}hh_{ddd} + \alpha_{2,3}h^2h_{ddd} + \alpha_{3,3}h^3h_{ddd} \big),
\end{aligned}
\tag{1}
$$

where $h_t = \frac{\partial h(d,t)}{\partial t}$, $h_d = \frac{\partial h(d,t)}{\partial d}$, $h_{dd} = \frac{\partial^2 h(d,t)}{\partial d^2}$, and $h_{ddd} = \frac{\partial^3 h(d,t)}{\partial d^3}$. We also note that $g$ is always zero in all PDEs, i.e., $g(d, t; h) = 0$.

In many cases, it is hard to solve the forward problem and hence general purpose PDE solvers do not exist. Nevertheless, one can use the following optimization to train a neural network $f(d, t; \boldsymbol{\theta})$ to approximate the solution function $h(d, t)$ as shown in Figure 2 (Raissi et al., 2019):

$$
\arg\min_{\boldsymbol{\theta}} \ L_I + L_B + L_G,
\tag{2}
$$

$$
L_I \stackrel{\text{def}}{=} \frac{1}{N_I} \sum_d \big( f(d, 0; \boldsymbol{\theta}) - h(d, 0) \big)^2,
\tag{3}
$$

$$
L_B \stackrel{\text{def}}{=} \frac{1}{N_B} \sum_{(d_{bc}, t)} \big( f(d_{bc}, t; \boldsymbol{\theta}) - h(d_{bc}, t) \big)^2,
\tag{4}
$$

$$
L_G \stackrel{\text{def}}{=} \frac{1}{N_G} \sum_{(d,t)} g(d, t; f, \boldsymbol{\theta})^2,
\tag{5}
$$

where $N_I, N_B, N_G$ are the numbers of training samples, $L_I$ is to train $\boldsymbol{\theta}$ for the initial condition, $L_B$ is for the boundary condition, and $L_G$ is for the governing equation. Because the governing equation is always zero, we simply minimize its squared term. Note that i) $f_t, f_d, f_{dd}, f_{ddd}$ can be easily constructed using the automatic differentiation implemented in TensorFlow or PyTorch, and ii) we only need $h(d, 0), h(d_{bc}, t)$, which are known a priori, to train the parameters $\boldsymbol{\theta}$.

## 2.2 INVERSE PROBLEM OF PDEs IN GENERAL CONTEXTS

The inverse problem is to find a governing equation given i) an initial condition $h(d, 0)$ and ii) a solution function $h(d, t)$ (Raissi, 2018). It learns $\alpha_{i,j}$ in Eq. 1 with the following loss (if possible, they use reference solutions as well):

$$
\arg\min_{\alpha_{i,j}} \frac{1}{N_G} \sum_{(d,t)} g(d, t; h)^2.
$$

Given a solution function $h$ and its partial derivative terms, we train $\alpha_{i,j}$ by minimizing the objective loss. Note that we know $h$ in this case. Therefore, the objective loss is defined with $h$ rather than with $f$, unlike Eq. 5.

The optimal solution of $\alpha_{i,j}$ is not unique sometimes. However, we note that no trivial solutions, e.g., $\alpha_{i,j} = 0$ for all $i, j$, exist for the inverse problem.

## 3    PDE-REGULARIZED NEURAL NETWORKS

Our goal in this work is to replace a system of ODEs (cf. Figure 1 (a)) with a PDE. Assuming that a target task-specific PDE is known *a priori*, given an initial condition $\boldsymbol{h}(0)$ extracted by the feature extractor from a sample $x$, a forward problem can be solved via the method described in Section 2.1. However, a target task-specific PDE is not known *a priori* in general, and thus, the governing equation should be learned from data via solving the inverse problem. Unfortunately, the solution function $h(d, t)$ is not also known *a priori* in our setting. Therefore, we make an assumption on the governing equation that it consists of the most common partial derivative terms (cf. Eq. 1) and then we propose to solve the forward and the inverse problems alternately: to train $\boldsymbol{\theta}$, we fix its governing equation $g$ (more precisely, $\alpha_{i,j}$ for all $i, j$), and to train $\alpha_{i,j}$ for all $i, j$, we fix $\boldsymbol{\theta}$.

**How to Solve Forward Problem.**    We customize the method presented in Section 2.1 by i) adding a task-specific loss, e.g., cross-entropy loss for image classification, ii) parameterizing the neural network $f$ by the initial condition $\boldsymbol{h}(0)$, and iii) dropping the boundary condition. Let $f(\boldsymbol{h}(0), d, t; \boldsymbol{\theta})$ be our neural network to approximate $h(d, t)$ given the varying initial condition $\boldsymbol{h}(0)$[1]. The definition of the governing equation is also extended to $g(d, t; f, \boldsymbol{h}(0), \boldsymbol{\theta})$. We use the following loss definition to train $\boldsymbol{\theta}$:

$$\underset{\boldsymbol{\theta}}{\arg\min} \quad L_T + \hat{L}_I + \hat{L}_G, \tag{6}$$

$$\hat{L}_I \overset{\text{def}}{=} \frac{1}{N_X} \sum_{x \in X} \Big( \frac{1}{\dim(\boldsymbol{h})} \sum_{d} \big( f(\boldsymbol{h}(0), d, 0; \boldsymbol{\theta}) - h(d, 0) \big)^2 \Big), \tag{7}$$

$$\hat{L}_G \overset{\text{def}}{=} \frac{1}{N_X} \sum_{x \in X} \Big( \frac{1}{N_H} \sum_{(d,t) \in H} g(d, t; f, \boldsymbol{h}(0), \boldsymbol{\theta})^2 \Big), \tag{8}$$

where $L_T$ is a task-specific loss, $X$ is a training set, and $H$ is a set of $(d, t)$ pairs, where $d \in \mathbb{R}_{\geq 0}, t \in \mathbb{R}_{\geq 0}$, with which we construct the hidden vector that will be used for downstream tasks, denoted by $\boldsymbol{h}^{\text{task}}$ (See Figure 3).

We query $f(\boldsymbol{h}(0), d, t; \boldsymbol{\theta})$ with the $(d, t)$ pairs in $H$ to construct $\boldsymbol{h}^{\text{task}}$. One more important point to note is that in order to better construct $\boldsymbol{h}^{\text{task}}$, we can train even the pairs in $H$ as follows: $\arg\min_{(d,t) \in H} L_T$ (line 7 in Alg. 1). Thus, the elements of $\boldsymbol{h}^{\text{task}}$ can be collected from different dimensions and layers. A similar approach to optimize the end time of integral was attempted for neural ODEs in (Massaroli et al., 2020).

**How to Solve Inverse Problem.**    After fixing $\boldsymbol{\theta}$, we train $\alpha_{i,j}$ for all $i, j$ by using the following $L_1$ regularized loss with a coefficient $w$:

$$\underset{\alpha_{i,j}}{\arg\min} \quad \hat{L}_G + R_G, \tag{9}$$

$$R_G \overset{\text{def}}{=} w \sum_{i,j} |\alpha_{i,j}|. \tag{10}$$

We minimize the sum of $|\alpha_{i,j}|$ to induce a sparse governing equation according to Occam's razor and since in many PDEs, their governing equations are sparse. This optimization allows us to choose a sparse solution among many possible governing equations. In many cases, therefore, our regularized inverse problem can be uniquely solved.

---

[1]Therefore, one can consider that our neural network $f$ approximates a general solution rather than a particular solution. A general solution means a solution of PDE with no specified initial conditions and a particular solution means a solution of PDE given an initial condition. Both neural ODEs and PR-Net approximate general solutions because initial conditions are varied.

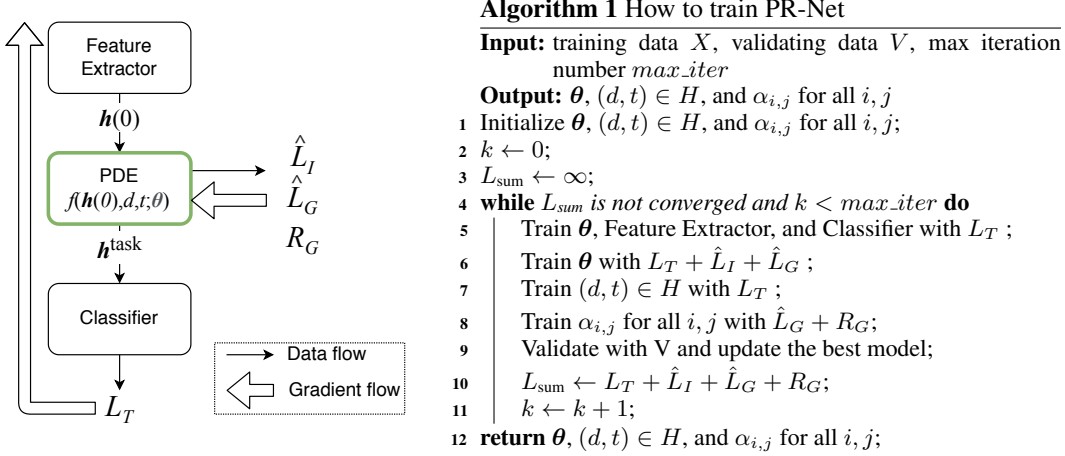

**Algorithm 1** How to train PR-Net

**Input:** training data $X$, validating data $V$, max iteration number $max\_iter$
**Output:** $\theta$, $(d, t) \in H$, and $\alpha_{i,j}$ for all $i, j$
1 Initialize $\theta$, $(d, t) \in H$, and $\alpha_{i,j}$ for all $i, j$;
2 $k \leftarrow 0$;
3 $L_{\text{sum}} \leftarrow \infty$;
4 **while** $L_{sum}$ *is not converged and* $k < max\_iter$ **do**
5      Train $\theta$, Feature Extractor, and Classifier with $L_T$ ;
6      Train $\theta$ with $L_T + \hat{L}_I + \hat{L}_G$ ;
7      Train $(d, t) \in H$ with $L_T$ ;
8      Train $\alpha_{i,j}$ for all $i, j$ with $\hat{L}_G + R_G$;
9      Validate with V and update the best model;
10      $L_{\text{sum}} \leftarrow L_T + \hat{L}_I + \hat{L}_G + R_G$;
11      $k \leftarrow k + 1$;
12 **return** $\theta$, $(d, t) \in H$, and $\alpha_{i,j}$ for all $i, j$;

Figure 3: The general architecture and the training algorithm of PR-Net

**Training Algorithm.** Our overall training algorithm is in Alg. 1. We alternately train $\theta$, $(d, t) \in H$, and $\alpha_{i,j}$ for all $i, j$. The forward problem to train $\theta$ becomes a well-posed problem (i.e., its solution always exists and is unique) if the neural network $f$ is analytical or equivalently, uniformly Lipschitz continuous (Chen et al., 2018). Many neural network operators are analytical, such as softplus, fully-connected, and exponential. Under the mild condition of analytical neural networks, therefore, the well-posedness can be fulfilled. The inverse problem can also be uniquely solved in many cases due to the sparseness requirement. As a result, our proposed training algorithm can converge to a cooperative equilibrium. Note that $\theta$, $(d, t) \in H$, and $\alpha_{i,j}$ for all $i, j$ cooperate to minimize $L_T + \hat{L}_I + \hat{L}_G + R_G$. Therefore, the proposed training method can be seen as a cooperative game (Mas-Colell, 1989). After finishing the training process, $\alpha_{i,j}$, for all $i, j$, are not needed any more (because $\theta$ already conforms with the learned governing equation at this point) and can be discarded during testing.

For complicated downstream tasks, training for $L_T$ should be done earlier than others (line 5). Then, we carefully update the PDE parameters (line 6) and other training procedures follow. The proposed sequence in Alg. 1 produces the best outcomes in our experiments. However, this sequence can be varied for other datasets or downstream tasks.

**Complexity Analyses.** The adjoint sensitivity method of neural ODEs enables the space complexity of $\mathcal{O}(1)$ while calculating gradients. However, its forward-pass inference time is $\mathcal{O}(\frac{1}{s})$, where $s$ is the (average) step-size of an underlying ODE solver. Because $s$ can sometimes be very small, its inference via forward-pass can take a long time.

Our PR-Net uses the standard backpropagation method to train and its gradient computation complexity is the same as that in conventional neural networks. In addition, the forward-pass inference time is $\mathcal{O}(1)$, given a fixed network $f$, because we do not solve integral problems.

## 4 EXPERIMENTS

In this section, we introduce our experimental evaluations with various datasets and tasks. All experiments were conducted in the following software and hardware environments: UBUNTU 18.04 LTS, PYTHON 3.6.6, NUMPY 1.18.5, SCIPY 1.5, MATPLOTLIB 3.3.1, PYTORCH 1.2.0, CUDA 10.0, and NVIDIA Driver 417.22, i9 CPU, and NVIDIA RTX TITAN. In Section J of Appendix, we summarize detailed dataset information and additional experiments.

Table 2: Image classification in MNIST and SVHN. The inference time is the time in seconds to classify a batch of 1,000 images. In general, PR-Net shows the best efficiency.

| Name | # Params | MNIST | | SVHN | |
|---|---|---|---|---|---|
| | | Test Accuracy | Inference Time | Test Accuracy | Inference Time |
| ResNet | 0.60M | 0.9966 | 7.6447 | **0.9660** | **8.6721** |
| RK-Net | 0.22M | 0.9970 | 7.4774 | 0.9652 | 13.5139 |
| ODE-Net | 0.22M | 0.9964 | 24.8355 | 0.9599 | 37.6776 |
| PR-Net | 0.21M | **0.9972** | **6.5023** | 0.9615 | 9.8263 |

## 4.1 IMAGE CLASSIFICATION WITH MNIST AND SVHN

We reuse the convolutional neural network, called ODE-Net, in the work by Chen et al. (2018) to classify MNIST and SVHN and replace its ODE part with our proposed PDE, denoted PR-Net in Table 2. See Appendix for the architecture and the hyperparameters of the network $f$ in PR-Net for this experiment. We reuse their codes and strictly follow their experimental environments.

Its detailed results are summarized in Table 2. We compare with ResNet, RK-Net and ODE-Net. In ResNet, we have a downsampling layer followed by 6 standard residual blocks (He et al., 2016). For RK-Net and ODE-Net, we replace the residual blocks with an ODE but they differ at the choice of ODE solvers. RK-Net uses the fourth-order Runge–Kutta method and ODE-Net uses the adaptive Dormand–Prince method for their forward-pass inference — both of them are trained with the adjoint sensitivity method which is a standard backward-pass gradient computation method. Our PR-Net, which does not require solving integral problems, shows the best performance in all aspects for MNIST. In particular, PR-Net shows much better efficiency than ResNet, considering their numbers of parameters, i.e., 0.60M of ResNet and 0.21M of PR-Net. Comparing ODE-Net and PR-Net for the inference time, our method shows much faster performance, i.e., 24.8355 seconds of ODE-Net vs. 6.5023 seconds of PR-Net to classify a batch of 1,000 images. Considering its short inference time, in SVHN we can say that its efficiency is still better than that of ODE-Net. One interesting point is that using the fourth-order Runge–Kutta method in RK-Net produces better accuracy and inferente time than ODE-Net in our experiments, which is slightly different from the original neural ODE paper (Chen et al., 2018). We tested more hyperparameters for them.

## 4.2 IMAGE CLASSIFICATION WITH TINY IMAGENET

We use one more convolutional neural network to test with Tiny ImageNet. Tiny ImageNet is the modified subset of ImageNet with downscaled image resolution $64 \times 64$. It consists of 200 different classes with 100,000 training images and 10,000 validation images. Our baseline model is Isometric MobileNet V3 (Sandler et al., 2019). For the efficient nature of ODE-Net and PR-Net, we consider that the resource-scarce environments, for which MobileNet was designed, are one of their best application areas. The isometric architecture of Isometric MobileNet V3 maintains constant resolution throughout all layers. Therefore, pooling layers are not needed and computation efficiency is high, according to their experiments. In addition, neural ODEs require an isometric architecture, i.e., the dimensionality of $\boldsymbol{h}(t)$, $t \geq 0$, cannot be varied. In our PR-Net, we do not have such restrictions. For fair comparison, however, we have decided to use Isometric MobileNet V3. We replace some of its MobileNet V3 blocks with ODEs or PDEs, denoted ODE-Net and PR-Net in Table 3, respectively. We train our models from scratch without using any pretrained network, with a synchronous training setup.

Table 3 summarizes their results. We report both of the top-1 and the top-5 accuracy, which is a common practice for (Tiny) ImageNet. In general, our PR-Net shows the best accuracy. PR-Net achieves an top-1 accuracy of 0.6157 with 4.56M parameters. The full Isometric MobileNet V3 marks an top-1 accuracy of 0.6578 with 20M parameters and the reduced Isometric MobileNet V3 with 4.30M parameters shows an top-1 accuracy of 0.6076. Considering the large difference on the number of parameters, PR-Net's efficiency is high. In particular, it outperforms others in the top-5 accuracy by non-trivial margins, e.g., 0.7911 of ODE-Net vs. 0.8115 of Isometric MobileNet V3 vs. 0.8357 of PR-Net. In addition, PR-Net shows faster forward-pass inference time in comparison with ODE-Net. The inference time is to classify a batch of 1,000 images.

Table 3: Image classification in Tiny ImageNet. PR-Net shows better efficiency than ODE-Net.

| Name | M.Net V3 | ODE-Net | PR-Net | M.Net V3 | ODE-Net | PR-Net |
|---|---|---|---|---|---|---|
| Width Multiplier | 1 | 1 | 1 | 2 | 2 | 2 |
| Mobile Blocks | 4 | 3 | 3 | 4 | 3 | 3 |
| ODE Blocks | N/A | 1 | N/A | N/A | 1 | N/A |
| PDE Blocks | N/A | N/A | 1 | N/A | N/A | 1 |
| Accuracy (top-1) | 0.5809 | 0.5547 | **0.5972** | 0.6076 | 0.5672 | **0.6157** |
| Accuracy (top-5) | 0.8049 | 0.7946 | **0.8166** | 0.8115 | 0.7911 | **0.8357** |
| # Params | 1.21M | 1.36M | 1.36M | 4.30M | 4.90M | 4.56M |
| Inference Time | **4.14** | 5.26 | 5.23 | **5.21** | 8.3 | 6.25 |
| Out-of-distribution Robustness (top-1 accuracy) | | | | | | |
| Gaussian Noise | 0.4495 | 0.4165 | **0.4685** | 0.4757 | 0.4474 | **0.4878** |
| Random Crop & Resize | 0.4636 | 0.4305 | **0.4841** | 0.4814 | 0.4419 | **0.4965** |
| Random Rotation | 0.3961 | 0.3667 | **0.4267** | 0.4256 | 0.3901 | **0.4381** |
| Color Jittering | 0.4206 | 0.3812 | **0.4429** | 0.4555 | 0.4108 | **0.4693** |
| Out-of-distribution Robustness (top-5 accuracy) | | | | | | |
| Gaussian Noise | 0.68 | 0.6619 | **0.7064** | 0.7025 | 0.6757 | **0.7205** |
| Random Crop & Resize | 0.7106 | 0.6935 | **0.7357** | 0.7215 | 0.6936 | **0.7442** |
| Random Rotation | 0.6372 | 0.6216 | **0.6627** | 0.6546 | 0.6319 | **0.6778** |
| Color Jittering | 0.6742 | 0.6396 | **0.6878** | 0.6874 | 0.6506 | **0.713** |

## 4.3 EXPERIMENTS ON ROBUSTNESS WITH TINY IMAGENET

To check the efficacy of learning a governing equation, we conduct three more additional experiments with Tiny ImageNet: i) out-of-distribution image classification, ii) adversarial attacks, and iii) transfer learning to other image datasets. In the first and second experiments, we apply many augmentation/perturbation techniques to generate out-of-distribution/adversarial images and check how each model responses to them. Being inspired by the observations that robust models are better transferred to other datasets (Engstrom et al., 2019a; Allen-Zhu & Li, 2020; Salman et al., 2020), in the third experiment, we check the transfer learning accuracy to other image datasets. According to our hypothesis, PR-Net which knows the governing equation for classifying Tiny ImageNet should show better robustness than others (as seen in Figure 4 for a scientific PDE problem in Appendix).

Neural networks are typically vulnerable to out-of-distribution and adversarial samples (Shen et al., 2016; Azulay & Weiss, 2019; Engstrom et al., 2019b). As being more fitted to training data, they typically show lower robustness to out-of-distribution and adversarial samples. However, PR-Net's processing them should follow its learned governing equation. Therefore, one way to understand learning a governing equation is a sort of regularization which prevents overfitting and implanting knowledge governing the classification process.

**Out-of-Distribution Image classification.** We use four image augmentation methods: i) adding a Gaussian noise of $\mathcal{N}(0, 0.1)$, ii) cropping a ceter area by size $56 \times 56$ and resizing to the original size, iii) rotating into a random direction for 30 degree, and iv) perturbing colors through randomly jittering the brightness, contrast, saturation, and hue with a strength coefficient of 0.2. All these are popular out-of-distribution augmentation methods (Shen et al., 2016; Azulay & Weiss, 2019; Engstrom et al., 2019b).

Our PR-Net shows the best accuracy (i.e., robustness) in all cases. In comparison with ODE-Net, it shows much better robustness, e.g., 0.3812 of ODE-Net vs. 0.4429 of PR-Net for the color jittering augmentation. One interesting point is that all methods are commonly more vulnerable to the random rotation and the color jittering augmentations than the other two augmentations.

**Adversarial Attack Robustness.** It is well-known that neural networks are vulnerable to adversarial attacks. Because the governing equation regularizes PR-Net's behaviors, it can be robust to unknown adversarial samples. We use FGSM (Goodfellow et al., 2015) and PGD (Madry et al., 2018) to find adversarial samples and the robustness to them is reported in Table 4. With various settings for the key parameter $\epsilon$ that controls the degree of adversarial perturbations, we generate adversarial samples. The configuration of doubling the number of channels used in each layer, denoted as "Width Multiplier 2", showed better performance in Table 3 and we use only the con-

Table 4: Adversarial attacks in Tiny ImageNet. PR-Net shows better robustness than ODE-Net.

| Attack Method | M.Net V3 | ODE-Net | PR-Net | M.Net V3 | ODE-Net | PR-Net |
|---|---|---|---|---|---|---|
| | Top-1 accuracy | | | Top-5 accuracy | | |
| FGSM($\epsilon = 0.5/255$) | 0.3860 | 0.3656 | **0.4041** | 0.6492 | 0.6398 | **0.6911** |
| FGSM($\epsilon = 1/255$) | 0.2304 | 0.2287 | **0.2499** | 0.4751 | 0.4928 | **0.5374** |
| FGSM($\epsilon = 3/255$) | 0.0452 | **0.0464** | 0.0369 | 0.1232 | 0.1562 | **0.1596** |
| PGD ($\epsilon = 0.5/255$) | 0.3733 | 0.3525 | **0.3910** | 0.6508 | 0.6409 | **0.6936** |
| PGD ($\epsilon = 1/255$) | 0.1902 | 0.1908 | **0.2133** | 0.4579 | 0.4810 | **0.5281** |
| PGD ($\epsilon = 3/255$) | 0.0218 | **0.0235** | 0.017 | 0.0792 | 0.1093 | **0.1144** |

Table 5: Transfer learning in Tiny ImageNet. PR-Net shows better transferability than ODE-Net.

| Dataset | M.Net V3 | ODE-Net | PR-Net | M.Net V3 | ODE-Net | PR-Net |
|---|---|---|---|---|---|---|
| | Top-1 accuracy | | | Top-5 accuracy | | |
| CIFAR100 | 0.7676 | 0.7460 | **0.7750** | 0.9320 | 0.9270 | **0.9480** |
| CIFAR10 | 0.9403 | 0.9280 | **0.9418** | **0.9963** | 0.9928 | 0.9962 |
| Aircraft | 0.6233 | 0.6027 | **0.6612** | 0.8509 | 0.8300 | **0.8561** |
| Food-101 | 0.7317 | 0.7128 | **0.7366** | 0.9108 | 0.9036 | **0.9174** |
| DTD | 0.4819 | 0.4973 | **0.5113** | 0.7660 | 0.7465 | **0.7957** |
| Cars | **0.6313** | 0.5576 | 0.6283 | **0.8380** | 0.7998 | 0.8319 |

figuration for this adversarial attack and next transfer learning experiments. For all attacks except FGSM($\epsilon = 3/255$) and PGD ($\epsilon = 3/255$), PR-Net shows the best robustness as shown in Table 4. The gap between PR-Net and other baselines are significant for the most cases of PGD.

**Transfer Learning.** As reported in (Engstrom et al., 2019a; Allen-Zhu & Li, 2020; Salman et al., 2020), robust models tend to produce feature maps suitable for transfer learning than regular models. In this regard, we checked the transferability of the pretrained PR-Net for Tiny ImageNet to other datasets: CIFFAR100 (Krizhevsky, 2009), CIFAR10 (Krizhevsky, 2009), FGVC Aircraft (Maji et al., 2013), Food-101 (Bossard et al., 2014), DTD (Cimpoi et al., 2014), and Cars (Yang et al., 2015). As shown in Table 5, PR-Net shows the best transfer learning accuracy in all cases except Cars. The improvements over M.Net V3 and ODE-Net are significant for Aircraft and DTD.

## 5 DISCUSSIONS & CONCLUSIONS

It recently became popular to design neural networks based on differential equations. In most cases, ODEs are used to approximate neural networks. In this work, on the other hand, we presented a PDE-based approach to design neural networks. Our method simultaneously learns a regression model and a governing equation that conform with each other. Therefore, the internal processing mechanism of the learned regression model should follow the learned governing equation. One can consider that this mechanism is a sort of implanting domain knowledge into the regression model. The main challenge in our problem definition is that we need to discover a governing equation from data while training a regression model. Therefore, we adopt a joint training method of the regression model and the governing equation.

To show the efficacy, we conducted five experiments: i) MNIST/SVHN classification, ii) Tiny ImageNet classification, iii) classification with out-of-distribution samples, iv) adversarial attack robustness, and v) transfer learning. Our method shows the best accuracy and robsutness (or close to the best) only except SVHN. In particular, the challenging robustness experiments empirically prove why learning an appropriate governing equation is important.

One limitation on this method is that it is sometimes hard to achieve a good trade-off between all different loss and regularization terms. Our method intrinsically involves various terms and we found that it is important to tune hyperparameters (especially for various coefficients and learning rates) in order to achieve reliable performance. In particular, $\alpha_{i,j}$, for all $i, j$, are important to learn reliable governing equations. Because the trained network $f$ is greatly influenced by the governing equation, hyperparameters should be tuned to learn meaningful equations. We also plan to study the proposed concept for many other classification/regression tasks.

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
