# OpenReview forum: "PDE-regularized Neural Networks for Image Classification"
_ICLR.cc/2021/Conference — Reject_

### Official Review · AnonReviewer2 · 2020-10-22

**Rating:** 5
**Confidence:** 4

**Review:**

Post-discussion update: The authors gave a fantastic, thoughtful and exhaustive response that did clarify all of my concerns about the paper. They also updated the paper considerably (making much better), and crucially changed the title to be very accurate the contents.

I like the paper now a lot, but the unimpressive results still stand. The PDE-based image classification performs ok, but also sometimes does not work very well. This would still be ok if insightful analysis of why the model improves would be provided. Unfortunately there is almost none of this, and then the contribution is more in the engineering side than science.

I would not object acceptance, but I would prefer the work to be more complete in this regard first. I raise my score to 5.

----

The paper proposes to solve the forward and inverse problems of PDEs simultaneously (that is, learn both the governing differential, and the forward solution surrogate). This is a dramatic and bold idea, but the paper does not explain why this combination would be a good idea, or what’s the benefit. It’s unclear why is the solution surrogate useful. The experiments show that this provides good results on image classification, but the PDE motivation is lacking. The resulting model does not require any integration, which is a major advantage. However, the paper should be more transparent on discussing the disadvantages of lack of integrals. Without forward solutions, the model is at risk of cumulating errors over time.

The paper seems to borrow its ideas almost completely from Raissi2019, and differences to it needs to be explicated. It seems that this paper takes Raissi2019 method and adds a loss function suitable for image classification. Given that none of the experiments are actually about learning PDEs (they are all image classification), this paper is very misleadingly titled. The method is also very incremental, and seems more like an application of Raissi2019 than an independent research work.

It’s also difficult to see why one would use a PDE for image classification at all. Labelling small images is already effectively a solved problem. I fail to see the PDE'ness of images.

This work also does not actually learn a “neural PDE”, since the governing equations are assumed to be 16-parameter predefined function, and not a neural function. Only the solution surrogate seems to be a neural network. The title is then misleading also in this regard.

I also have hard time seeing why not develop this bidirectional method for ODEs first? The paper should do this as an ablation study to first show that it grants some benefits in the simpler ODE case.

The paper is written in a confusing manner, and lacks presentation polish (typos, language mistakes, strange figure order, lots of dubious statements, etc). The presented methods are also not introduced properly, and it seems that the reader needs intricate understanding of Raissi2019 first.

The experiments show that the PDE-net, ODE-net and ResNet are all equally good at classifying MNIST and SVHN (with no significant differences). The results are missing log-likelihoods, standard deviations and training time analytics. It’s difficult to see what’s the benefit of the method here. In Tiny-experiments the comparison target of MobileNet seems arbitrary. Why compare to a mobile phone -optimized classifier, given that PDE’s surely are far from ideal on such settings. There are no large-scale image classification tasks, nor standard image baseline methods (alexnet, vgg, wresnet). The ResNet comparison is also missing from Tiny.

The out-of-distribution experiments are excellent, and show the PDE’s improve clearly from ODEs and beat MobileNet. These results are very interesting, and potentially significant. Here more exhaustive experiments should be done, and comparisons to other augmentation methods performed. The authors should also try to give insight why the PDE is more robust to perturbations.

Overall the paper presents an incremental improvement to PDE learning with limited novelty, with unclear presentation, unclear motivation and mixed but partially promising results. The paper needs more work, and should be reworked to be more independent of Raissi2019, and refocused (incl. title) towards the promising application of image robustness, and *why* the PDE are more robust than ODEs in this setting.


Technical comments:
o I do not understand what the dimension “d” means. It does not seem to a dimension at all, but instead a state vector of the state space? The notation is very misleading
o Neural ODE’s do not have particularly small number of parameters (they are often applied in very simple cases or in small latent spaces, but here other NN’s would be simple as well)
o what is “procrastinate”?
o the paper confuses layers and time to be equivalent, this is not true in neural ODEs
o where is eq 1 coming from, and why is the model only restricted to 3rd order (monomial) differentials? Surely a more general PDE definition could have been used
o “general purpose PDE solvers do not exist”: surely they exist, but are too slow to be practical
o eq 6: what is “h”?
o eq 6: why is Raissi2019 performance studied here? I fail to see what’s the relevance of repeating someone else’s work. Is there some novelty here?
o fig4: all methods seem to have very poor fits, given that this is a simple 1D problem with massive amount of data.
o sec3: h(d,t) should be a function of “h0” as well, and its unclear if this is a true or proxy solution.

---

> ### Author Response · Authors · 2020-11-18
> **Continue to upload**
>
> 8. The symbol “d” means “dimension” of hidden vectors for neural networks (and corresponds to space in scientific PDE problems). The d-th element of the hidden vector at layer t is modeled by h(d,t) in our work whereas the entire hidden vector at layer (time) t is modeled by h(t) in neural ODEs.
>
> 9. It is true that conventional networks can also be compact. However, there are several successful works achieving state-of-the-art performance with an order of smaller number of parameters after adopting neural ODEs, e.g., [Pinckaers et al., Neural Ordinary Differential Equations for Semantic Segmentation of Individual Colon Glands, 2019].
>
> 10. The term “procrastinate” means delayed forward inference time of neural ODEs for solving integral problems (refer to Table 2 for the inference time).
>
> 11. The residual connection is the same as the explicit Euler method (as noted in the seminal neural ODE paper). Therefore, neural ODEs are to generalize residual connections by interpreting their layers (time) as continuous variables. So, h(t) in neural ODEs means the entire hidden vector of residual connections at layer (time) t.
>
> 12. Eq 1 is a popular dictionary of governing equations. They are popular terms in governing equations. In many cases, governing equations are defined as a combination of those 16 terms [Peng et al, Accelerating Physics-Informed Neural Network Training with Prior Dictionaries, 2020]. In particular, they are called a dictionary in this field.
>
> 13. By the sentence “general purpose PDE solvers do not exist”, we meant that in many PDEs, it is hard to solve their forward problems. In order to obtain accurate and scientifically meaningful solutions, problem-specific knowledge often needs to be hard-coded to the software implementation of solvers in an intrusive way. We will revise our manuscript to reflect our intention more precisely.
>
> 14. We will move Figure 4 to Appendix. Figure 4 is different from Raissi et al. (2019). Raissi et al. (2019) conducted experiments with interpolations. Our Figure 4 is extrapolation results to emphasize the importance of learning governing equations.
>
>
> 15. Figure 4 has poor predictions because they are all extrapolation results, which is different from Raissi et al. (2019). In Raissi et al. (2019), all interpolation results are excellent. For extrapolations, however, we found that the role of governing equations is much more important than that in interpolations. As reported, Figure 4 (b) didn’t forget about the existence of the valley around  x = 0, which is much better than Figure 4 (d). Our point is that even in the worst case, learning governing equations provides better conformance to underlying physical dynamics.
>
> 16. The initial vector h0 is actually the initial conditions of h(d,t) where t = 0.

---

> ### Author Response · Authors · 2020-11-18
> **Thanks for your comments.**
>
> 0. A revised manuscript and a revised supplementary material will be uploaded soon. We will let you know again after uploading them.
>
> 1. We believe the forward computations (as well as the backward computations) with time-stepping algorithms would be more prone to accumulating errors over time. As reported in (Zhung et al, Adaptive Checkpoint Adjoint Method for Gradient Estimation in Neural ODE, ICML 2020), adaptive ODE solvers, such as DOPRI, sometimes produce underflow errors for their step sizes, i.e., too large errors keep decreasing the step sizes and eventually they become smaller than the smallest precision of the IEEE 754 floating point standard.
>
> 2-1. Raissi et al. (2019) propose two separate methods: 1) a method for solving forward problems (i.e., computing solutions for given PDEs) and 2) a method for solving inverse problems (i.e., identifying PDEs given solution measurements). Raissi et al. (2019) ``never’’ solved these two separate problems at the same time.
>
> 2-2. Ruthotto and Haber (2019) proposed PDE-inspired networks to achieve stable forward computation for systems of ODEs derived from e.g., symplectic integration schemes developed for Hamiltonian systems.
>
> 2-3. To our knowledge, Long et al. (2018) is the only one solving the inverse and the forward PDE problems at the same time. But, their setting is to solve scientific PDE problems whereas we solve for downstream tasks. So, their method is rather straightforward in comparison with our method and their setting does not include any machine learning task-specific loss. They do not guarantee equilibrium solutions either. Our work requires 1) a carefully devised solution surrogate neural network, 2) a sophisticated mechanism to train it. To our knowledge, we are the first to solve such complex problems for downstream machine learning tasks.
>
> 3. PDE has long been studied and utilized as a tool in many computer visions and image processing tasks, e.g., dating back to 1990s, textbook edge-detection algorithm [Perona and Malik, TPAMI, 1990], total-variation-based noise removing [Rudin, Osher, Fatemi, Physica D, 1992], and recently, PDE-inspired deep learning architectures for image classification  [Ruthotto and Haber, 2019, Haber, et al, AAAI, 2019], and image denoising [Jia, et al, CVPR, 2019].
>
> 4-1. The solution surrogate h is a neural network in our case. So, its partial derivative terms in the governing equation, such as u_x, u_t, u_xx, and so forth, are computed using automatic differentiation and then construct a computational graph (i.e., a neural network) as a linear combination of the computed partial derivatives. This type of approach (i.e., attaching automatic differentiation as a part of the forward pass of neural networks) has appeared in many papers including Hamiltonian neural networks [Greydanus et al, 2019].
>
> 4-2. The rationale for having 16 prespecified partial derivatives is that they are the most common terms that appear in many PDEs [Peng et al, Accelerating Physics-Informed Neural Network Training with Prior Dictionaries, 2020]. In particular, they are called a dictionary in this field.
>
> 5. Our model does "learn" PDEs consisting of many partial derivatives from image data. This is an analogous setting to neural ODEs: where the model is designed as a system of ODEs, and the model learns the dynamics as a form of a parameterized velocity function u_t = f(u), where f is a parameterized function (e.g., neural networks). The applications of neural ODEs are not restricted to learning scientific ODEs but can be applied to image classification, continuous normalizing flows, and many other downstream tasks. In this regard, our paper title can be considered as an homage to the seminal neural ODE paper, titled “Neural Ordinary Differential Equations.”
>
> 6. Large-scale neural networks for image classification already show superhuman performance. However, neural networks under resource scarcity still require more studies. We aim to devise a neural network with small memory requirements and, thus, chose ODE-Net and MobileNet as our baseline to compare. We made this more clear in the manuscript.
>
> 7. We have added more experiments per the reviewer's suggestion: robustness to adversarial attacks with FGSM and PGD and transfer learning from Tiny Imagenet to 6 other image datasets. For all those additional experiments, the proposed neural PDEs outperformed the baselines in almost all cases. We also added training loss curve charts, training time analyses, robustness analyses with t-SNE visualization of hidden vectors, and feature map analyses. We believe that our additional materials significantly improve the quality of the discussion. We will upload a revised manuscript and supplementary material soon. You can check soon.

---

> ### Author Response · Authors · 2020-11-25
> **Uploaded new version.**
>
> Thanks for your concrete comments. Following your suggestions, we changed the paper title and many other things. We also removed the unclear statement "procrastinate the forward-pass inference" and explicitly said that "the forward-pass inference can sometimes take a long time." Our key modifications are highlighted in red. We also added many more experiments in the main paper and in the supplementary material.

---

### Official Review · AnonReviewer3 · 2020-10-25
**The paper is an reasonable extension to the idea of neural ODEs. By treating both layer and the hidden dimensions as continuous variables, the proposed method alternately solve the regression model and governing equations that conform to each other, without solving integral problems in neural ODEs. The experiment results showed a good accuracy comparing with neural ODEs.**

**Rating:** 7
**Confidence:** 4

**Review:**

The main contribution of this paper is to address the numerical instability issue in neural ODE method when solving the integral problems. To avoid the integral problems, the new methods treat both the layer and hidden dimensions as continuous variable, and solve them at the same time by learning a regression model.

I also like the idea of learning governing equation and regression model together by an alternating algorithm, which in theory should be better than training a differential equation based neural network with priori knowledge of governing equations.

My main concern is that to avoid the integral problem in neural ODE, the authors paid the price to treat the whole neural network as a fully coupled system and had to solve for all the layers simultaneously, which needs more variables/parameters to tune. In addition, there is also a number of parameters to add in order to solve for governing equations. All this changes will make the hyper-parameter tuning much more difficult than neural ODEs. The authors should comment on this point during the rebuttal period.

The new proposed method did give a new insight to handle the numerical instability in neural ODEs, with a bit cost of making model more complicated to tune though. Overall, I would recommend a weakly accept.

---

> ### Author Response · Authors · 2020-11-18
> **Thanks for your comments.**
>
> A revised manuscript and a revised supplementary material will be uploaded soon. We will let you know again after uploading them.
>
> Hyper-parameter tuning is indeed more difficult than those in other types of neural networks. Therefore, we devised a sophisticated training mechanism (Algorithm 1) in our work, which is one of our main contributions. The sparsity of parameters alpha_{i,j} is also important in our work. We recommend users control them at the beginning. In addition, it is also very important to adopt a reasonable architecture for the solution surrogate neural network h. If an ill-defined neural network h is used, it is much more difficult to train with. For instance, the feature extractor of MNIST uses a series of convolutions followed by ReLU and there are no negative values in the initial condition. In such a case, we also use ReLU in the solution surrogate neural network h. By that, it is much easier to train for initial conditions.
>
> All in all, it is true that neural PDEs are harder to design and train than conventional neural networks. However, we believe that other users can follow our guidance to stabilize their tasks after some moderate amount of customization.

---

> ### Author Response · Authors · 2020-11-25
> **Upload new version.**
>
> Thanks for your concrete comments. Following Reviewer2's suggestions, we changed the paper title and many other things. We also removed the unclear statement "procrastinate the forward-pass inference" and explicitly said that "the forward-pass inference can sometimes take a long time." Our key modifications are highlighted in red. We also added many more experiments in the main paper and in the supplementary material.

---

### Official Review · AnonReviewer4 · 2020-10-28
**Some novelty in the method; more experiments are required**

**Rating:** 6
**Confidence:** 4

**Review:**

Summary:
The paper proposed the method of neural PDE as an improvement of neural ODE. In specific, neural PDE considers both the layer and the hidden dimension as continuous variables of the PDE. The new part of neural PDE compared to neural ODE is essentially solving PDE inverse problems (learning PDE from data) in the computational mathematics and engineering community, and the way of learning PDE (by embedding the PDE and initial condition into the loss function via automatic differentiation) is the physics-informed neural network (PINN) proposed in [Raissi et al., JCP, 2019]. The experiments show that compared to neural ODE, neural PDE achieves comparable accuracy but with less forward-pass inference time; but these experiments are not convincing enough.

pros:
- Because the proposed method uses automatic differentiation to handle the derivative as PINN and thus can avoid the numerical integration, and the inference time of neural PDE is less than that of neural ODE.

Major comments:

Results:
- The paper only shows the results on small problems in two tables, which is convincing. To show the performance of this method, the authors should add more experiments and also show more details of the behavior of the method, e.g., training.
- The training procedure is complicated. It includes 4 steps: first train L_T, then L_T + L_C + L_G, etc. It is not clear why the authors train in this way. Are the hyperparameters difficult to tune? How stable is the training?
- The loss includes high-order derivatives of network f, which makes the training much more expensive. What is the computational cost of training? Also, the training trajectories should be added to show the convergence behavior of the loss.

Methods:
- The novelty of neural PDE is replacing ODE in neural ODE with PDE, and adds the PDE loss. But the way of handling PDE loss is exactly the PINN [Raissi et al., JCP, 2019], which is one of the main references in the paper. The authors should state this clearly.
- The authors use the example of Allen-Cahn equation to deliver the message that training with PDE would have better accuracy for extrapolation. However, this example of Allen-Cahn equation is exactly the same example used in [Raissi et al., JCP, 2019]. It might be OK to repeat the example and result from another paper to deliver their message, but the authors should state it clearly. Also, this phenomenon has already been observed in the computational engineering community, e.g., https://www.biorxiv.org/content/10.1101/865063v2 . In fact, I think this part is not the main part of the paper, and can be moved into appendix.
- Section 3 is not well written and the description is not clear.
    - (d,t) are the “symbolic” variables of PDE, and the user can choose their values arbitrary to compute the loss L_G. But they are also the parameters to be trained. Are the (d,t) in L_G the same as the (d,t) used for construct h_last?
    - Is (d, t) the same for all the inputs?
    - Is d in one dimension?
    - It is not clear how the authors select H? How many (d,t) pairs are in H?
    - Are d and t unbounded?
    - It seems that the authors use some points (d,t) in the whole domain to construct h_last; if this is the case, then the “last” is not correct, which usually means the PDE solution at the last time.
- In appendix, why the input dim of f is 6^2x67 not 6^2x67+2? because there are extra inputs of d and t. Why does the network f use ReLU? The derivative f_{dd} would be zero everywhere. The authors should show the details of the network including the (d,t) part.

Minor comments:
- In the introduction, it is not correct that “they studied PDEs in scientific problem domains and do not consider t as a continuous variable but use a set of discretized points of t.” In PINN, t is treated as the continuous variable. The authors should be aware of this.
- Some notations are confusing. For example, d is usually the dimension, but here d is the space variable.
- Why the authors use L_C to denote that initial condition? Why not L_I? since L_B is for “boundary” and L_G is for “governing”.
- The appendix does not have all the information of hyperparameters, e.g., what is the size of H?

---

> ### Author Response · Authors · 2020-11-18
> **Continue to upload.**
>
> 15-1. We use a feature map size of 6x6x67 in order to feed (d,t) pairs as additional channels. The first 64 channels are from the feature extractor and the last 3 channels are added by us. The feature map is a 3D data structure but for efficiency, we discretize its last dimension. Therefore, d is in R^2 (instead of R^3 because we discretize the last dimension). We use d_1 and d_2 to denote the first and second dimensions of d, respectively. Then, we need a channel for each of d_1, d_2, and t after forgetting the discretized dimension d_3. The original 64 channels + 3 additional channels becomes 6x6x67. In other words, all elements in the same position of those 64 channels share the same values for d_1, d_2, and t. This increases the efficiency of our method.
>
> 15-2. Since the feature extractor of MNIST and SVHN uses a series of convolutions followed by ReLU, we also use ReLU in the solution surrogate neural network h. By that, it is much easier to train for initial conditions. At the same time, the solution h(d,t) cannot be negative if assuming a convolutional neural network with a ReLU at the end, which is our model for which we learn a governing equation.

---

> ### Author Response · Authors · 2020-11-18
> **Thanks for your comments.**
>
> 1. A revised manuscript and a revised supplementary material will be uploaded soon. We will let you know again after uploading them.
>
> 2. We have added more experiments per the reviewer's suggestion: robustness to adversarial attacks with FGSM and PGD and transfer learning from Tiny Imagenet to 6 other image datasets. For all those additional experiments, the proposed neural PDEs significantly outperformed the baselines. We also added training loss curve charts, training time analyses, robustness analyses with t-SNE visualization of hidden vectors, and feature map analyses. We believe that our additional materials significantly improve the quality of the discussion. We will upload a revised manuscript and supplementary material soon. You can check soon.
>
> 3. Our training mechanism is sophisticatedly devised considering the difficulty of solving the inverse and the forward problems at the same time. However, we found L_T is still the most important. Sacrificing L_T to stabilize PDE-related loss values was not a sensible decision for our work. Therefore, the starting point is always training L_T first in our work. Then, we train for PDE-related loss functions. Then, we train (d,t) pairs in H again before training alpha_{i,j} values because the task performance is important, as the set H of (d,t) pairs are also important.
>
> 4. In modern deep learning platforms, the automatic differentiation (the Jacobian-vector-product) is very well supported without explicitly calculating the Jacobian matrix. Therefore, our training complexity is not as high as it seems. We will upload new training overhead analyses soon.
>
> 5. By the sentence “they studied PDEs in scientific problem domains and do not consider t as a continuous variable but use a set of discretized points of t.”, we meant Long et al. (2018). We will make it clear.
>
> 6. We will change L_C to L_I to denote the loss for initial conditions.
>
> 7-1. We will make it clear about the connection between our work and PINN. Raissi et al. (2019) propose two separate methods: 1) a method for solving forward problems (i.e., computing solutions for given PDEs) and 2) a method for solving inverse problems (i.e., identifying PDEs given solution measurements). Raissi et al. (2019) ``never’’ solved these two separate problems at the same time.
>
> 7-2. Ruthotto and Haber (2019) proposed PDE-inspired networks to achieve stable forward computation for systems of ODEs derived from e.g., symplectic integration schemes developed for Hamiltonian systems.
>
> 7-3. To our knowledge, Long et al. (2018) is the only one solving the inverse and the forward PDE problems at the same time. But, their setting is to solve scientific PDE problems whereas we solve for downstream tasks. So, their method is rather straightforward in comparison with our method and their setting does not include any machine learning task-specific loss. They do not guarantee equilibrium solutions either. Our work requires 1) a carefully devised solution surrogate neural network, 2) a sophisticated mechanism to train it. To our knowledge, we are the first to solve such complex problems for downstream machine learning tasks.
>
> 8. We will move the Allen-Cahn equation result to Appendix as recommended. We also agree on the point.
>
> 9. We use a set of (d,t) pairs in H for L_G. From the set H of (d,t) pairs, we also construct h_last.
>
> 10. We learn only one set H of (d,t) pairs. They are uniformly initialized at the beginning and we optimize them during our proposed training procedures.
>
> 11. The symbol “d” means the dimension of the hidden vector (or the space in scientific PDEs).
>
> 12. The cardinality of H is the same as the dimensionality of the neural ODE state vector in a certain dataset for a fair comparison. This also shows an advantage of our work. In neural ODEs, we can control only t, but in neural PDEs we can control both d and t.
>
> 13. The values for d and t have no limitations theoretically (as t can also be very large in neural ODEs). In practice, however, we found that they varied in a reasonable bound, e.g., a lower bound around 0 and an upper bound around some small value, and did not diverge in experiments. However, this characteristic can be different for other tasks and datasets, which is a future research direction.
>
> 14. After reading the comment, we found that the term “last” is misleading. We used the term because it corresponds to the last hidden vector of conventional neural networks used for classification. We will instead use the term “task-oriented vector.” Since the elements in h_last can come from all different d and t, we believe the new term “task-oriented vector” is a good choice.

---

> ### Author Response · Authors · 2020-11-25
> **Upload new version.**
>
> Thanks for your concrete comments. Following Reviewer2's suggestions, we changed the paper title and many other things. We also removed the unclear statement "procrastinate the forward-pass inference" and explicitly said that "the forward-pass inference can sometimes take a long time." Our key modifications are highlighted in red. We also added many more experiments in the main paper and in the supplementary material.

---

### Official Review · AnonReviewer1 · 2020-10-28

**Rating:** 6
**Confidence:** 3

**Review:**

1. Summary:

The authors propose Neural PDE as an enhanced alternative of Neural ODE, which learns both the governing equations and the target labels by alternating between solving the forward problem and the backward problem.

2. Clearly state your recommendation (accept or reject):

I am leaning towards recommending an accept to this paper, as it proposes a coherent way of solving supervised-learning tasks with Neural PDE.

3. Strong points:

The model is reasonable. Unlike Neural ODE, it does not require numerical integration. It outperforms models including ResNet and ODE-Net in image classfication tasks, and can also generalize to out-of-distribution samples.

4. Ask questions you would like answered by the authors:

a) The main question I have is the relationship to prior work, including Ruthotto and Haber (2019), Long et al. (2018) and Raissi et al. (2019). The authors did mention, for example, that the main difference with Long et al. (2018) is that the latter focuses on scientific problems and only uses a set of discretized points of t. What else are the novelties compared to the previous approaches, besides these as well as alternating between the training of the forward problem and that of the backward problem? It may also be reasonable to compare against those models in the experiments.

b) Any reasons why the boundary conditions removed, and how is the set H of (d, t) pairs selected?


5 Additional comments:

On the topic of physics-informed differential-equations-based models, there are some other works that may be worth referencing:
[1] Greydanus, Samuel, Misko Dzamba, and Jason Yosinski. "Hamiltonian neural networks."
[2] Chen, Zhengdao, Jianyu Zhang, Martin Arjovsky, and Léon Bottou. "Symplectic recurrent neural networks."
[3] Cranmer, Miles, Sam Greydanus, Stephan Hoyer, Peter Battaglia, David Spergel, and Shirley Ho. "Lagrangian neural networks."
[4] Zhong, Yaofeng Desmond, Biswadip Dey, and Amit Chakraborty. "Symplectic ode-net: Learning hamiltonian dynamics with control."

---

> ### Author Response · Authors · 2020-11-18
> **Thanks for your comments.**
>
> 1. A revised manuscript and a revised supplementary material will be uploaded soon. We will let you know again after uploading them.
>
> 2. Raissi et al. (2019) propose two separate methods: 1) a method for solving forward problems (i.e., computing solutions given PDEs) and 2) a method for solving inverse problems (i.e., identifying PDEs given solution measurements). Raissi et al. (2019) "never’’ solved these two separate problems at the same time.
> Ruthotto and Haber (2019) proposed PDE-inspired networks to achieve stable forward computation for systems of ODEs derived from e.g., symplectic integration schemes developed for Hamiltonian systems.
>
> 3. To our knowledge, Long et al. (2018) is the only one solving the inverse and the forward PDE problems at the same time. But, their setting is to solve scientific PDE problems whereas we solve for downstream tasks. So, their method is rather straightforward in comparison with our method and their setting does not include any machine learning task-specific loss. They do not guarantee equilibrium solutions either. Our work requires 1) a carefully devised solution surrogate neural network, 2) a sophisticated mechanism to train it. To our knowledge, we are the first to solve such complex problems for downstream machine learning tasks.
>
> 4. We model the dynamics of neural networks as neural PDEs. Unlike PDEs that arise in natural science, it is non-trivial to define the boundary conditions in our problem settings. Inspired by PDEs with no boundary conditions, we model neural PDEs with no boundary conditions and have demonstrated empirically that there is no issue for not having the boundary condition. Nonetheless, we acknowledge that the boundary conditions of neural PDEs are worth studying, in particular, periodic boundary conditions.
>
> 5. The set H of (d,t) pairs are also subject to be trained in our work as mentioned in the paper. They are uniformly initialized at the beginning and we optimize them during our proposed training procedures.
>
> 6. We are aware of those works mentioned by the reviewer. We haven’t included those papers in our original submission as those papers can be considered as extensions of the neural ODEs by incorporating, for example, Hamiltonian structure, symplectic architectures, and so on, and our intention in this paper is to suggest an alternative to the family of neural ODEs. However, during the revision, as per the reviewer’s suggestion, we thought it would be more informative to provide a complete list of papers for the family of neural ODEs and those papers are now cited in the newer version.

---

> ### Author Response · Authors · 2020-11-25
> **Uploaded new version.**
>
> Thanks for your concrete comments. Following Reviewer2's suggestions, we changed the paper title and many other things. We also removed the unclear statement "procrastinate the forward-pass inference" and explicitly said that "the forward-pass inference can sometimes take a long time." Our key modifications are highlighted in red. We also added many more experiments in the main paper and in the supplementary material.

---

### Decision · Program_Chairs · 2021-01-07
**Final Decision**

**Decision:**

Reject

**Comment:**

This paper proposes a method for regularizing image classifiers by encouraging their hidden activations to conform to a PDE.  This is a reasonable idea, and the authors clearly improved the paper a lot in response to the reviews.  However, the main tasks of MNIST and SVHN classification seem way too easy, and the baselines all need to be tuned to be as fast as possible for a given accuracy, if that's the relevant metric.  I agree with the reviewers that this line of work is promising but that the current paper is not sufficiently illuminating or well-executed to meet ICLR standards.